# Towards Principled Design for Graph Neural Networks Through Governing Law of Dynamic Learning Behavior

## Abstract

Graph Neural Networks (GNNs) have been extensively evaluated in the machine learning regime. In contrast to most studies that primarily focus on the empirical assessment of model performance across different graph datasets, we ratchet the gear of GNN benchmarking another notch forward to understand the graph learning mechanism that shapes characteristic learning behaviors of each GNN model. Specifically, we introduce a comprehensive benchmark framework PDEGNN-Bench to evaluate GNNs derived from six representative governing equations, i.e., partial differential equations (PDEs), for graph heat isotropic/anisotropic diffusion, non-local diffusion, reaction–diffusion, Hamiltonian system, wave transport, and oscillatory synchronization. By linking each GNN model instance to its corresponding governing equation, we establish new insights into the design principle for new GNNs by understanding the relationship between mechanistic interpretations and descriptive learning performance. To that end, we seek to explore two fundamental questions: (1) How well does each governing equation respond to the challenge of over-smoothing in GNNs? (2) How does the homogeneity degree of graph topology influence model performance across PDE families? Taken together, our benchmark provides a systematic evaluation of leading GNN models through the lens of underlying physical mechanisms. Through well-designed experiments, we demonstrate that each family of governing equations exhibits distinct model generalization and interpretability characteristics, offering guidance for designing suitable GNNs for the new graph data.

## 1 Introduction

Inspired by graph theory and deep learning, Graph neural networks (GNNs) have emerged as a powerful framework for representation learning on complex structured data (Kipf & Welling, 2016), revolutionizing fields such as drug discovery (Stokes et al., 2020) and brain connectomics (Parisot et al., 2018; Zhou et al., 2020; Wu et al., 2020; Bronstein et al., 2017). Particularly, recent advancements have introduced a rich variety of physics-inspired GNN architectures governed by partial and ordinary differential equations (PDE/ODE), which inject explicit inductive biases rooted in dynamical systems theory. PDE-governed GNNs (Eliasof & Treister, 2021; Poli et al., 2023) have gained attention due to their well-founded theoretical formulations, the capacity to model continuous-time dynamics and enhanced model interpretability. By enabling a principled manipulation of graph propagation processes, such as energy preservation and wave transport, these models provide a flexible framework for tailoring message-passing behavior to the underlying structure of the data.

Despite their theoretical elegance and modeling versatility of PDE-governed GNNs, the current literature reveals substantial gaps in both theoretical understanding and empirical evaluation. Methodology surveys (Han et al., 2023; Liu et al., 2025) primarily emphasize theoretical formulations or investigate families in limited settings, but fall short of delivering comprehensive quantitative comparisons across diverse dynamical equations. At the same time, most existing benchmarks (Dwivedi et al., 2023; Hu et al., 2020) concentrate on architectural variants or application-specific models, while overlooking the rich physical and mathematical diversity offered by governing equations of dynamical systems.

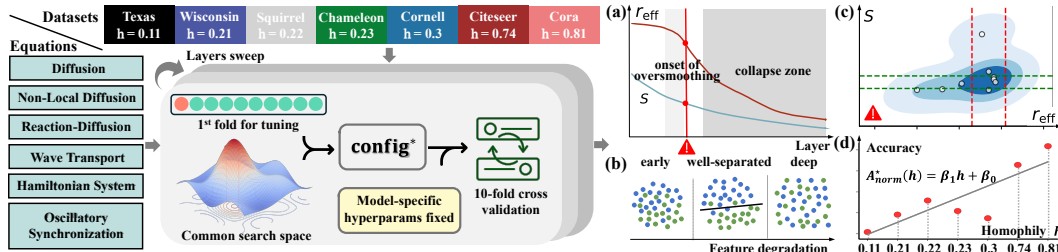

Figure 1: An overview of PDEGNN-BENCH for evaluating the dynamics in PDE-informed GNNs. PDEGNN-BENCH covers six classes of governing equations and seven datasets with varying homophily, with all models tuned in a common search space and evaluated under 10-fold cross-validation (Sec. 4.1). (a) The degradation of layer-wise features is monitored via two interpretable metrics: *effective rank* $r_{\text{eff}}$ and *class-mix score* $S$, where a sharp drop in either indicates the onset of oversmoothing (Sec. 4.2.1). (b) Visual illustration of class separation at early, intermediate, and deep layers, showing how oversmoothing leads to blurred class boundaries. (c) An envelope-based alarm system is constructed from the empirical distribution of $(r_{\text{eff}}, S)$ across models and depths, providing dataset-specific alarms for detecting oversmoothing (Sec. 4.2.2). (d) Model sensitivity to graph homophily, quantified by the slope $\beta_1$ from regression of accuracy against homophily (Sec. 4.3).

This oversight is particularly critical as PDE-GNNs are increasingly applied to domains such as chemistry, traffic forecasting, and neuroimaging, where graph homophily varies widely and model depth selection directly impacts both accuracy and runtime efficiency. In this context, two fundamental challenges emerge: oversmoothing, the degradation of representations with increasing depth, and homophily sensitivity, the dependence of model performance on structural alignment. Existing studies mainly focus on simple diffusion processes and evaluate smoothing effects on highly homophilous graphs (Zhao & Akoglu, 2019), there has been no systematic effort to examine they manifest across distinct PDE families, nor to analyze them jointly. Therefore, what remains unclear is how different governing dynamics shape the coupled evolution of depth-induced oversmoothing and topology-induced homophily effects. This gap is particularly critical in scientific domains such as brain network modeling, where both representation stability and structural interpretability are essential but remain poorly understood in relation to PDE-based dynamics.

To bridge these gaps in theory and evaluation, we propose PDEGNN-BENCH, the first comprehensive benchmark that systematically evaluates representative PDE/ODE-driven GNNs. As illustrated in Figure 1, we focus on six governing equations that (i) are widely studied in the literature (Han et al., 2023), (ii) admit closed-form continuous generators, and (iii) collectively span the spectrum from purely diffusive to fully conservative dynamics: isotropic/anisotropic diffusion, non-local diffusion, reaction–diffusion, Hamiltonian system, wave transport, and oscillator synchronization. PDEGNN-BENCH evaluates these models on seven datasets that cover homophily $h \in [0.11, 0.81]$ and differ in scale, sparsity, and label balance. By incorporating unified indicators for oversmoothing and homophily sensitivity, we provide a robust and reproducible evaluation framework.

The technical contributions in this work are highlighted as follows:

- **First cross-equation benchmark with oversmoothing and homophily diagnostics.** PDEGNN-BENCH provides the first systematic benchmark comparing six PDE-governed GNNs under unified training and evaluation protocols. Through in-depth analysis of depth-wise feature expressiveness and class separability (*effective rank* captures latent dimensional collapse, while *class-mix score* quantifies inter-class overlap, together they diagnose distinct aspects of oversmoothing that accuracy alone cannot reveal) as well as performance under varying homophily, our benchmark reveals distinct oversmoothing behaviors and homophily sensitivity, while also uncovering unique strengths and failure modes across PDE families.

- **Prior-driven, model-agnostic oversmoothing alarm.** Leveraging the validation optima of existing models, we learn a compact, dataset-specific prior that outlines the representation regime in which learning remains stable. This prior can act as a model-agnostic early-warning signal, automatically flagging and capping depth once feature quality begins to deteriorate during training, thereby enabling label-efficient, model-agnostic control of oversmoothing.

- **Quantitative assessment of homophily sensitivity.** We systematically analyze how structural homophily influences generalization by regressing normalized peak accuracy against homophily levels across various PDE-GNNs. The resulting slopes of regression offer interpretable measures of PDE-GNNs' reliance on homophilic structure. This unified protocol reveals that even theoretically robust PDE-informed GNNs exhibit varying degrees of homophily sensitivity in practice.

Taken together, these advantages position PDEGNN-BENCH as a unified framework through which both depth scalability and structural robustness of PDE-governed GNNs can be rigorously examined. By disentangling the roles of governing equations, learning behavior, and graph topology, our benchmark provides a reproducible foundation for future model design, automated depth control, and theory-driven analysis of graph representations.

## 2 RELATED WORK

Continuous–time perspectives on graph representation learning have attracted strong research attention. We briefly review the most relevant survey efforts and summarize related work.

**Surveys on Differential Equation-Inspired GNNs.** Han et al. (2023) provides the first systematic review of PDE-driven GNNs. They trace message passing back to discrete heat diffusion and categorize subsequent extensions as anisotropic, oscillatory, non-local, reaction-diffusion, Hamiltonian, etc. Liu et al. (2025) provides a broader *Graph NDE* survey covering ODEs, SDEs and PDEs. It is deliberately conceptual: it stresses modelling principles and open mathematical questions (e.g. higher-order derivatives, numerical solvers). Their taxonomy is task-oriented (classification, forecasting, generation) and emphasizes the roles a GNN can play inside a neural DE pipeline.

**Structural Limitations in GNNs.** Rusch et al. (2023) formalize oversmoothing as layer-wise exponential decay of a node-similarity functional and compares Dirichlet energy, MAD, and higher-order norms as diagnostics. It further reviews mitigation techniques such as PairNorm and DropEdge and highlights the risk of losing expressivity when merely constraining smoothness. At the same time, a growing body of work explores why certain oversmoothing metrics succeed or fail under varying structural homophily. Yan et al. (2022) shows that nodes with high heterophily and large relative degree oversmooth long before Dirichlet energy vanishes, revealing a blind spot of energy-based diagnostics. Capacity metrics such as effective rank (Zhang et al., 2025) track performance better in that regime.

While existing surveys map the rapidly expanding design space of differential-equation-inspired GNNs, no supply of empirical comparisons reveals how different governing equations behave on a uniform suite of datasets. Our work fills this gap by offering the *first quantitative benchmark* that instantiates representative models from each governing-equation family under identical training protocols, evaluating over smoothing dynamics and homophily sensitivity.

## 3 FRAMEWORK TAXONOMY: GOVERNING EQUATIONS IN GNN MODELS

Let $\mathcal{G} = (\mathcal{V}, \mathcal{E}, A, X)$ be a graph with $N = |\mathcal{V}|$ nodes, adjacency $A \in \mathbb{R}^{N \times N}$, and node features $X \in \mathbb{R}^{N \times d}$. The node states at time $t$ are $U(t) \in \mathbb{R}^{N \times d}$ (aka. graph feature representations with dimension $d$), with initial condition $U(0) = X$. We group existing PDE-informed GNNs into six coherent classes, each characterized by a prototypical evolution equation.

### 3.1 GRAPH DIFFUSION (PARABOLIC HEAT FLOW)

Diffusion-based models simulate heat propagation on a graph, governed by parabolic PDEs with *isotropic* or *anisotropic* conductance. In continuous form, node features $u_i(t) \in \mathbb{R}^d$ evolve as $\partial_t u_i(t) = \sum_j \nabla \cdot \big( A_{ij} \, \alpha_{ij}(t) \, \nabla u_j(t) \big)$, where $A_{ij}$ is the fixed edge weight, $\alpha_{ij}(t) \in [0, 1]$ is a conductance (constant, stochastic, or learned), and $\nabla, \nabla \cdot$ are the discrete gradient and divergence.

**Local & Isotropic Diffusion.** With $\alpha_{ij} \equiv 1$, the operator reduces to the graph Laplacian $\Delta = D - A$, $D_{ii} = \sum_j A_{ij}$, so that the dynamics become $\partial_t u_i(t) = -\sum_j \Delta_{ij} u_j(t)$. This homogeneous diffusion underlies *GCN* (Kipf & Welling, 2016) and *GCNII* (Chen et al., 2020b), the latter augmenting above equation with identity mixing of initial features.

**Anisotropic or Feature-Adaptive Diffusion.** Allowing $\alpha(t)$ to vary introduces $\partial_t u_i(t) = -\sum_j \Delta_{ij}^{\alpha(t)} u_j(t)$, where $\Delta^{\alpha(t)} := D^{\alpha(t)} - A \odot \alpha(t)$ and $D_{ii}^{\alpha(t)} = \sum_j A_{ij}\alpha_{ij}(t)$. This feature-driven Laplacian defines geometry-adaptive diffusion. *GRAND* (Chamberlain et al., 2021) samples $\alpha(t)$ stochastically and integrates with a neural ODE solver, while *GAT* (Veličković et al., 2018) sets $\alpha_{ij}(t) = \mathrm{softmax}_j(\phi(u_i, u_j))$, adapting conductance via attention.

### 3.2 Non-Local Diffusion

Classical diffusion acts locally on edges, which limits expressive power on graphs with long-range interactions. *fLode* (Maskey et al., 2023) introduces fractional powers of the Laplacian: $\partial_t u_i(t) = -\sum_j (-\Delta)_{ij}^s u_j(t), 0 < s < 1$, with $(-\Delta)^s = U\Lambda^s U^\top$. The power-law kernel induces virtual long-range edges, capturing strong heterophily in a single step by creating virtual edges that link far-apart nodes.

### 3.3 Reaction–Diffusion Dynamics

Reaction–diffusion (RD) systems couple Laplacian smoothing with nonlinear reactions, enabling spatial pattern formation and richer expressivity. For Laplacian $\Delta$ and kinetics $F$, $\partial_t u_i(t) = -\sum_j \Delta_{ij}u_j(t) + F(u_i(t))$. *GREAD* (Choi et al., 2023) uses Fisher–KPP kinetics $F(u) = \beta u(1-u)$, yielding stable patterns that resist collapse. *ACMP* (Wang et al., 2022) extends to two species $(u, v)$ with Allen–Cahn dynamics, $\partial_t u_i = -\sum_j \Delta_{ij}u_j + u_i - u_i^3 - v_i$ and $\partial_t v_i = -D_v \sum_j \Delta_{ij}v_j + \varepsilon(u_i - \kappa)$, where $v$ acts as an inhibitor, producing cluster-like phase separation.

### 3.4 Hamiltonian System Dynamics

Certain graph processes, such as molecular motion or power-grid oscillations, are conservative and naturally described by Hamiltonian mechanics. The state vector $\mathbf{z}(t) = (\mathbf{q}(t), \mathbf{p}(t)) \in \mathbb{R}^{2dN}$ collects node-wise positions $\mathbf{q}$ and momenta $\mathbf{p}$, $\dot{\mathbf{z}}(t) = \mathbf{J}\nabla_{\mathbf{z}}\mathcal{H}(\mathbf{z}; A), \mathbf{J} = \begin{pmatrix} 0 & I \\ -I & 0 \end{pmatrix}$, where $\mathcal{H}(\mathbf{z}; A)$ is the Hamiltonian energy functional. And $\mathbf{J}$ defines the canonical skew-symmetric matrix. This flow preserves both volume and total energy $\mathcal{H}$. *HamGNN* (Kang et al., 2023) parameterizes $\mathcal{H}$ with a neural network and integrates above equation via symplectic solvers.

### 3.5 Wave Dynamics

Diffusion suppresses high-frequency modes, whereas hyperbolic wave equations propagate them at finite velocity, periodically exchanging kinetic and potential energy: $\partial_{tt}u_i(t) + c^2 \sum_j \Delta_{ij}u_j(t) = 0$, where $c > 0$ is the wave speed. This inductive bias complements diffusion by transmitting oscillatory components along edges. *GraphCON* (Rusch & Mishra, 2022) generalizes above equation with damping and nonlinear forcing: $\partial_{tt}u_i + \alpha\,\partial_t u_i + c^2 \sum_j \Delta_{ij}u_j(t) = \sigma\left(\sum_j \Delta_{ij}u_j(t)\right), \alpha > 0$, where $\alpha$ dissipates energy and $\sigma(\cdot)$ provides a learnable band-pass filter. The resulting dynamics combine wave propagation, controlled attenuation, and nonlinear expressivity.

### 3.6 Oscillatory Synchronization

Beyond diffusion or waves, networks of coupled oscillators exhibit collective synchronization. *KuramotoGNN* (Nguyen et al., 2024) treats each node feature $u_i(t)$ as a phase evolving under the classical Kuramoto model: $\partial_t u_i(t) = \omega_i + K\sum_j a_{ij}\sin(u_j(t) - u_i(t))$, with learnable natural frequency $\omega_i$, global coupling $K > 0$, and attention-derived weights $a_{ij} = \mathrm{softmax}_j((W_K u_i^0)^\top (W_Q u_j^0)/\sqrt{d_k})$. Large $a_{ij}$ drive phase locking, while weaker couplings sustain modular rhythms, balancing global synchrony and community diversity.

*BRICK* (Ding et al., 2025) instead augments the dynamics with adaptive control and geometry-aware constraints: $\partial_t u_i = \omega_i + \gamma\,\Pi_{u_i}\left(y_i + \sum_j (W \odot A)_{ij}u_j\right)$, where $\omega_i = \xi_\sigma(x_i)$ and $y_i = \zeta_\mu(x_i)$ are learned from the initial feature $x_i$, $W$ is a symmetric trainable coupling, and $\Pi_u(z) = z - \langle z, u\rangle u$

projects onto the tangent space of the unit sphere. This introduces a geometry-consistent, physics-informed framework capable of modeling modular synchrony in graphs.

## 4 EXPERIMENTAL DESIGN AND EVALUATION FRAMEWORK

To systematically evaluate the behavior of GNNs governed by different PDE-based dynamics, we construct a unified experimental benchmark centered on three core questions: (1) how resilient is each model to oversmoothing as depth increases; (2) can we define a model-agnostic alarm system that detects such collapse in real time; and (3) how sensitive is each governing equation to structural homophily. This section outlines the proposed design for each question.

### 4.1 HYPERPARAMETER SEARCH PROTOCOL AND EXPERIMENTAL SETTINGS

We use seven datasets spanning homophily $h \in [0.11, 0.81]$ (Texas=0.11, Wisconsin=0.21, Squirrel=0.22, Chameleon=0.23, Cornell=0.30, Citeseer=0.74, Cora=0.81). For datasets with reported optimal hyperparameters in the original papers, we keep them unchanged and only vary the number of layers. For others, we use default arguments and perform tuning within a *common search space* of learning rate $\eta$ over a 3-point grid $\eta \in \{10^{-2}, 5 \times 10^{-3}, 10^{-3}\}$, while fixing weight decay to $10^{-4}$ and hidden dimension to $64$ to prevent overfitting and simplify the search space. All experiments are conducted under the 10-fold cross-validation splits from Geom-GCN (Pei et al., 2020), and the full hyperparameter list for all models and datasets is provided in our code. To investigate behavior across depth, for models without explicit layer definitions, we follow the strategy from GRAND (Chamberlain et al., 2021): we fix the ODE solver step size to 1, use a Runge–Kutta-4 integrator, and vary the integration time from 2 to 128 to simulate depths. In addition, we adopt a two-stage protocol that retains statistical rigor while cutting the wall-clock cost: the first split is used to select $\eta^\star$ over the depth grid, which is then fixed for the remaining runs.

### 4.2 OVERSMOOTHING BEHAVIOR UNDER VARIOUS GOVERNING EQUATIONS

#### 4.2.1 EXPERIMENTAL PROTOCOL AND RESULTS FOR GOVERNING–EQUATION OVERSMOOTHING-HOMOPHILY

A meaningful study must disentangle three orthogonal factors: 1. *Governing operator* (diffusion, reaction–diffusion, Hamiltonian, wave, oscillation). 2. *Graph homophily* $h \in [0.11, 0.81]$. 3. *Depth-induced degeneration* measured by specific measurements. Previous works typically fix one axis and vary another (e.g., MADGap (Chen et al., 2020a) on homophilous graphs, or Dirichlet energy on a single backbone), but none explore the full 3-D space. We therefore propose the following design. The analysis uses two complementary indicators (rationale detailed in App. A.1), computed layer-wise on the validation split:

**Effective rank** quantifies the intrinsic dimensionality of the node feature matrix at layer $\ell$. Let $\sigma_k^{(\ell)}$ be the $k$-th singular value of $X^{(\ell)} \in \mathbb{R}^{N \times d}$. The *effective rank* (Zhang et al., 2025) is expresses as:

$$r_{\text{eff}}^{(\ell)} = \exp\left[-\sum_{k=1}^{d} p_k^{(\ell)} \log p_k^{(\ell)}\right], \quad p_k^{(\ell)} = \sigma_k^{(\ell)} / \sum_{j=1}^{d} \sigma_j^{(\ell)},$$

A large $r_{\text{eff}}^{(\ell)}$ implies that the singular-value spectrum is relatively flat, so information is spread across many independent directions. A monotone drift toward $r_{\text{eff}} \approx 1$ indicates the spectrum collapses and all energy concentrates in its leading component, thus serving as a direct signature of representation collapse, i.e. oversmoothing in the form of rank-deficiency.

**Class–mix score** tracks how distinguishable the classes remain after layer $\ell$. Letting $\rho^{(\ell)} = E_{\text{w}}^{(\ell)}/(E_{\text{b}}^{(\ell)} + \varepsilon)$ (Yan et al., 2022) with within-class energy $E_{\text{w}}^{(\ell)}(X) = \sum_{y_i = y_j} \|x_i - x_j\|_2^2$ and between-class energy $E_{\text{w}}^{(\ell)}(X) = \sum_{y_i \neq y_j} \|x_i - x_j\|_2^2$ computed over the labeled validation nodes, we define a graph homophily-free normalization as $S^{(\ell)} = |\rho^{(\ell)} - 1|$. If oversmoothing homogenises node representations, the within–class energy $E_{\text{w}}$ and between–class energy $E_{\text{b}}$ become equal, it forces $\rho^{(\ell)} = E_{\text{w}}/E_{\text{b}} \to 1$ and thus $S^{(\ell)} \to 0$.

Together, the two metrics serve as complementary indicators of oversmoothing: one signals global rank collapse, while the other signals the loss of class separability.

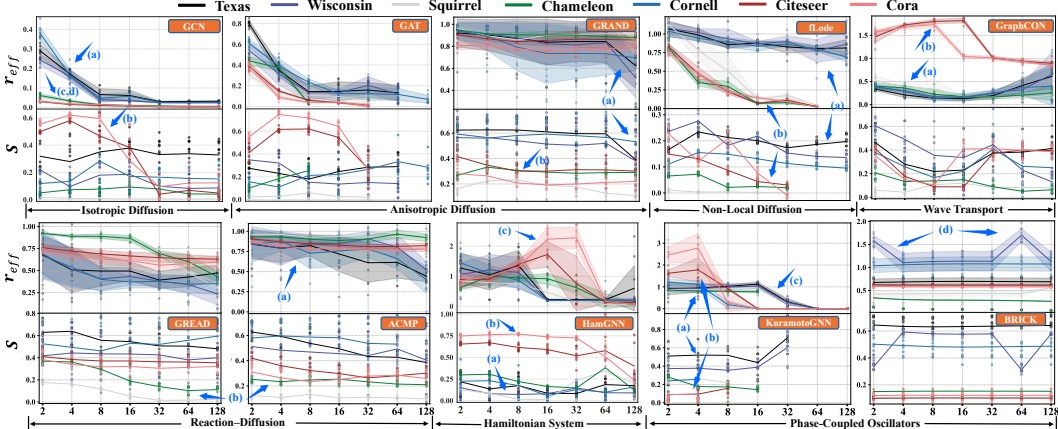

(a) Depth-wise trends of effective rank $r_{\text{eff}}$ and class-mix score $S$ across ten models and seven datasets (ordered by graph homophily). A decline in either metric indicates the onset of oversmoothing.

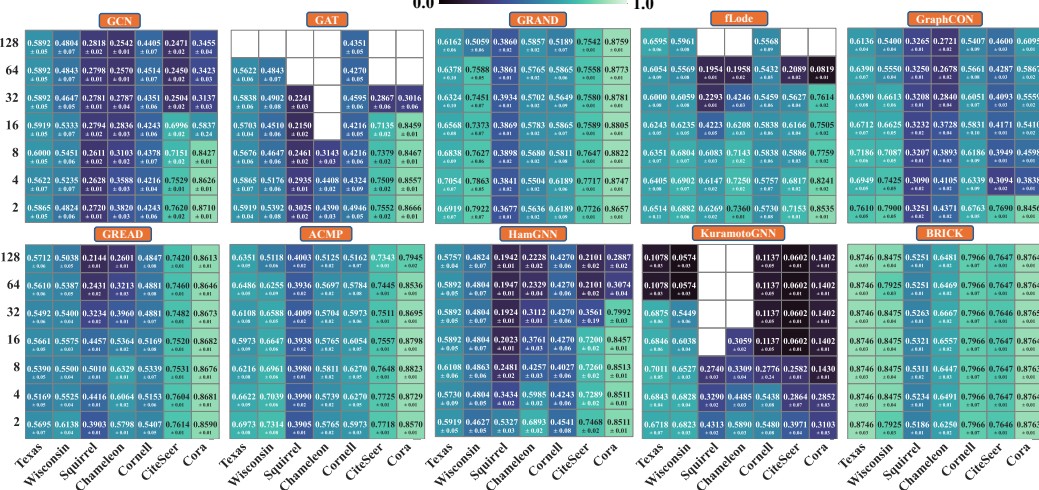

(b) Test accuracy (mean ± std) on seven datasets ordered by increasing homophily (left to right) and depth (bottom to top). Missing entries indicate unavailable or invalid results.

Figure 2: Overview of oversmoothing indicators and performance across depths and homophily.

**Joint trends in oversmoothing and accuracy across depth and homophily**

**From the perspective of governing equation.** Figure 2a presents the depth-wise evolution of effective rank $r_{\text{eff}}$ and class-mix score $S$, while Figure 2b shows corresponding test accuracy for ten models across seven datasets ordered by increasing homophily (from Texas to Cora). Declines in $r_{\text{eff}}$ or $S$ indicate reduced expressiveness or class separability, signaling potential oversmoothing. A summary of oversmoothing trends and tipping depths for each model class is provided in Table 1.

Isotropic Diffusion. *Discussion:* (a) GCN rapidly loses discriminative power after 4–8 layers on high-homophily graphs, as seen in (b) the sharp drop of $S$ (and accuracy) beyond 16 layers on Cora/Citeseer (Figures 2a, 2b). For low-homophily graphs, (c) both $S$ and accuracy stay low but stable, indicating limited capacity largely affected by depth. Meanwhile (d) $r_{\text{eff}}$ stays stable on high-homophily graphs, indicating preserved feature diversity despite failed classification. *Explanation:* isotropic diffusion (heat equation) uniformly damps high-frequency components across layers, leading to feature homogenization and oversmoothing.

Anisotropic Diffusion. *Discussion:* GRAND (a) maintains stable $r_{\text{eff}}$ and $S$ across depths, with degradation only on a few low-homophily graphs (e.g., Wisconsin, Texas) at 128 layers. Ac-

Table 1: Oversmoothing behavior summary based on layer-wise indicators $r_{\text{eff}}$ and $S$.

| Model Type | Low $h$ | High $h$ | Tipping Depth (low/high $h$) | Risk |
|---|---|---|---|---|
| Isotropic Diffusion (GCN) | Rapid collapse of $r_{\text{eff}}$, sharp $S$ peak | Collapse slightly delayed | 2–4 | **High** |
| Anisotropic Diffusion (GRAND) | Stable $S$ and $r_{\text{eff}}$ | Mostly stable; slight drop at deep layers | 64 / − | **Low** |
| Non-local Diffusion (fLode) | Mostly, gradual decline in $S$, minor drop in $r_{\text{eff}}$ | Noticeable $S$ drop, moderate $r_{\text{eff}}$ degradation | 8 / 4 | **Medium** |
| Wave-based Transport (GraphCON) | Smooth $r_{\text{eff}}$, stable up to moderate depth | Abrupt collapse at shallow depth | 16 / 2 | **Medium** |
| Reaction-Diffusion (GREAD) | Moderate degradation by 8 layers | High $r_{\text{eff}}$ but low $S$ persists | 8 / − | **Medium** |
| Reaction-Diffusion (ACMP) | Stable with depth, strong $r_{\text{eff}}$ until 128 layers | Mild drop in $S$ and $r_{\text{eff}}$ | 64 / − | **Low** |
| Hamiltonian (HamGNN) | Stable up to 4 layers | Irregular $r_{\text{eff}}$, collapse after 16 layers | 4 / 16 | **High** |
| Phase-coupled Oscillatory (KuramotoGNN) | Stable when solvable, some numerical failure | Failed $r_{\text{eff}}$ and $S$ at shallow layers | − / − | **Very High** |
| Phase-coupled Oscillatory (BRICK) | Stable across depths, minor drop in $S$ | Stable with high $S$, low degradation | − / − | **very Low** |

curacy tracks this stability, and (b) notably $S$ is higher on low- than high-homophily graphs, unlike other models. *Explanation:* Through noise injection and resampling, GRAND adaptively preserves local information. Its anisotropic diffusion adjusts edge weights to suppress irrelevant signals while preserving class boundaries.

Non-local Diffusion. *Discussion:* fLode (a) exhibits a decline in $r_{\text{eff}}$ and $S$ with depth, (b) more pronounced on high-homophily graphs where accuracy also lags behind competitors. *Explanation:* fractional diffusion enables non-local propagation but weakly preserves local class structure, causing subtle loss of discriminability and weaker $S$, despite only moderate reduction in feature diversity.

Wave-based Transport. *Discussion:* GraphCON (a) performs well on low-homophily graphs (e.g., Texas, Wisconsin), although both $r_{\text{eff}}$ and accuracy degrade gracefully with depth. However, on high-homophily graphs, (b) a sharp drop in $S$ and accuracy occurs at shallow depths (depth 4), revealing a sensitivity to over-depth stacking. *Explanation:* Although the wave equation's propagation mechanism maintains high-frequency signals, transmits information with finite speed, and supports long-range dependencies in low-homophily settings, excessive depth in homogeneous graphs may lead to phase interference and signal cancellation, undermining feature quality.

Reaction–Diffusion. *Discussion:* GREAD and ACMP remain stable with depth, though (a) ACMP generally achieves higher $r_{\text{eff}}$ and accuracy. Their trends align, (b) on low-homophily graphs (Squirrel, Chameleon), both yield low $S$ despite high $r_{\text{eff}}$, indicating features retain diversity but lack separability. *Explanation:* reaction–diffusion balances smoothing with local activations, preserving diversity but not discriminability unless reactions are class-specific.

Hamiltonian Systems. *Discussion:* HamGNN is (a) stable $S$ on low-homophily graphs (Texas, Wisconsin), with accuracy and $S$ maintained across depth. On high-homophily graphs (Citeseer), degradation starts at layer 8 with (b) collapsing $S$ and (c) irregular $r_{\text{eff}}$, and by layer 16 both accuracy and $r_{\text{eff}}$ drop sharply. *Explanation:* while Hamiltonian dynamics conserve energy, insufficient regularization causes chaotic trajectories as depth grows, and numerical errors and initialization amplify this, especially in high-homophily graphs.

Phase-coupled Oscillatory. *Discussion:* KuramotoGNN is (a) stable on low-homophily graphs but (b) struggles on high-homophily ones, where (c) numerical issues in $r_{\text{eff}}$ confirm instability, and reported results could not be reproduced. In contrast, BRICK remains robust across all datasets, (d) with only minor dips (e.g., Wisconsin) consistently reflected in $r_{\text{eff}}$, $S$, and accuracy. *Explanation:* phase coupling fosters synchrony while preserving oscillatory modes; BRICK adds stronger control and regularization, preventing collapse even at large depths.

**From the perspective of graph homophily and structural type.** On high-homophily graphs (e.g., Cora, Citeseer), anisotropic diffusion (GRAND), reaction–diffusion models (GREAD, ACMP), and phase-coupled oscillatory model (BRICK) remain stable across depth, showing strong resistance to oversmoothing. In contrast, isotropic diffusion (GCN), non-local diffusion (fLode), wave-based transport (GraphCON), and Hamiltonian systems (HamGNN) degrade rapidly with depth due to oversmoothing or collapse. On low-homophily graphs (Texas, Wisconsin, Squirrel, Chameleon, Cornell), models separate into two categories. Type I – Depth-stable / mild degradation: GCN (though with low accuracy), ACMP, and BRICK maintain stable behavior without severe oversmoothing. Type II – Heterophily-sensitive: performance varies with heterophily subtype. GRAND, GraphCON, and KuramotoGNN degrade on structurally heterophilous graphs (Texas, Wisconsin, Cornell), while fLode, GraphCON, GREAD, HamGNN, and KuramotoGNN fail on topologically heterophilous graphs (Squirrel, Chameleon). This highlights that robustness depends not only on depth but also on the type of heterophily. A summary of model performance trends categorized by graph homophily is provided in Table 2.

Table 2: Model performance trends categorized by graph homophily and structure.

| Graph Type | Stable Models | Oversmoothing / Degrading Models |
|---|---|---|
| High homophily (`Cora`, `Citeseer`) | GRAND, GREAD, ACMP, BRICK (resistant to oversmoothing) | GCN, fLode, GraphCON, HamGNN, KuramotoGNN (oversmooth or collapse with depth) |
| Low homophily (`Texas`, `Wisconsin`, `Squirrel`, `Chameleon`, `Cornell`) | **Type I – Depth-stable/mild degradation:** GCN (low accuracy), ACMP, BRICK **Type II – Heterophily-sensitive:** *Structural heterophily-sensitive (Texas, Wisconsin, Cornell):* GRAND, GraphCON, KuramotoGNN *Topological heterophily-sensitive (Squirrel, Chameleon):* fLode, GraphCON, GREAD, HamGNN, KuramotoGNN | |

**From the perspective of oversmoothing indicators.** Both *effective rank* ($r_{\text{eff}}$) and *class-mix score* ($S$) track accuracy (Acc) closely, which could serve as reliable proxies for oversmoothing. Abrupt drops (e.g., GCN within 8 layers, GRAND at 128 layers) or anomalies (e.g., a spike in $r_{\text{eff}}$ for HamGNN on `Cora` at 16 layers) consistently coincide with Acc degradation. Although numerical ranges vary across models, some patterns are clear: when both indicators remain very low (e.g., $< 0.1$ in GCN on `Chameleon`, `Squirrel`), the model typically fails to learn meaningful representations from the outset. When they remain stable at moderate-to-high levels (ACMP), the model demonstrates resistance to oversmoothing. Accordingly, Table 1 reports the tipping depth at which oversmoothing begins to degrade representations.

### 4.2.2 A MODEL–AGNOSTIC OVERSMOOTHING ALARM

The depth at which a GNN begins to oversmooth varies widely across models and datasets. But results reveal a common collapse geometry shared across models, thereby motivating a unified *oversmoothing alarm* whose signal is derived directly from hidden representations and thus provides a model-agnostic alternative to manual depth tuning.

Early heuristics monitor either global dispersion or pairwise distances. PairNorm (Zhao & Akoglu, 2019) normalizes the feature matrix to curb shrinkage but provides no stopping rule. MADGap (Chen et al., 2020a) triggers on a threshold of mean absolute deviation, but fails on strongly heterophilous graphs. DDCD (Shen et al., 2024) solely uses the difference $E_{\text{w}} - E_{\text{b}}$ as an early–stop proxy. To our knowledge, no prior work combines a capacity metric with a class–aware energy ratio into a single alarm. The mechanism below fills that gap.

**Oversmoothing alarm mechanism.** Given a set of validation–optimal quadruples $(\ell^{\star}, \text{Acc}_{\text{val}}^{(\ell^{\star})}, r_{\text{eff}}^{(\ell^{\star})}, S^{(\ell^{\star})})$ for every model–dataset pair (see the details in App. A.3), we collect $\mathcal{R}_d = \left\{ (r_{\text{eff},i}^{\star}, S_i^{\star}) \,|\, \text{record } i \text{ belongs to } d \right\}$ for each dataset $d \in \mathcal{D}$. We wish to extract a dataset–level prior that allows future practitioners to (i) locate the typical region where past architectures governed by different dynamic equations achieved their best trade-off between effective rank and class separability, and (ii) issue an early alarm when a new architecture drifts far from this region. To that end we transform the discrete samples into a $(r_{\text{eff}}, S)$ density heatmap and derive a 95%–level empirical envelope ($\mathcal{E}_d = \mathcal{I}_r^{(d)} \times \mathcal{I}_S^{(d)}$), which is detailed in App. A.4.

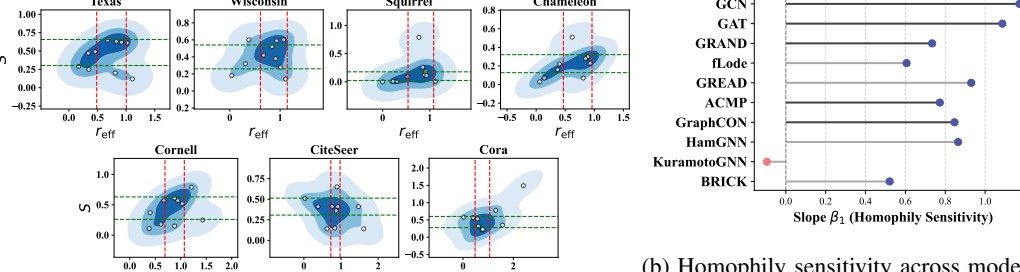

(a) Empirical $(r_{\text{eff}}, S)$ landscape for seven benchmark datasets. Grey dots indicate the validation–optimal operating points of ten models, progressively darker blue contours show higher levels of the bivariate KDE. Red (vertical) and green (horizontal) dashed lines show the dataset–specific 95% empirical envelope.

(b) Homophily sensitivity across models. Slopes $\beta_1$ from regressing validation accuracy on homophily $h$ ($Acc^{\star}(h) = \beta_0 + \beta_1 h$). Positive $\beta_1$ (blue) indicates benefit from homophily, while negative $\beta_1$ (red) suggests robustness to heterophily. Darker lines denote significant trends ($p < 0.05$).

Figure 3: Oversmoothing landscapes across datasets and homophily trends across models.

As illustrated in Figure 3a, across all seven datasets the $(r_{\mathrm{eff}}, S)$ density maps reveal a consistent qualitative pattern. Models whose optima fall inside this nucleus, e.g., the cluster around $(r_{\mathrm{eff}} \approx 0.7, \ S \approx 0.4)$ on `Wisconsin`, rarely display either rank collapse or excessive high-frequency noise, and therefore provide a reliable safe region for depth selection. By contrast, points outside the dashed envelope occur with probability below 5% and align with representation pathologies predicted by aforementioned conclusions. Specifically, those left of the envelope (low $r_{\mathrm{eff}}$), e.g., outliers on `Squirrel`, indicate severe oversmoothing, those above (high $S$), e.g., `Cora`'s solitary peak, reflect excessive high-frequency content without accuracy gain. In these cases, the validation curve soon plateaus or degrades, confirming the envelope as a *label-efficient early alarm*.

Algorithm 1 in App. A.4 details the procedure (with `Wisconsin` as an example) describes how to use this prior when training a new model. The key idea is to monitor $(r_{\mathrm{eff}}^{(\ell)}, S^{(\ell)})$ during depth growth and trigger early stop once the representation (i) leaves $\mathcal{E}_d$ and (ii) shows a persistent decline, which could indicate the onset of over-smoothing. It requires no hyperparameter search beyond loading the pre-computed envelope and is thus label–efficient. When validation labels are unavailable (e.g., in pre-training), the same logic can be applied with $r_{\mathrm{eff}}$ alone, with its lower envelope as the rank–collapse threshold.

Two heuristic principles guide this alarm: (i) If the representation lies within ranges of past successful models, this does not guarantee optimality, but the risk of rank collapse or high-frequency noise is low. (ii) If it deviates, the event serves as a warning. One should check validation curves, and if performance improves, add the new point to the experience set $\mathcal{R}_d$ and update the envelope, allowing the prior to adapt to genuine innovations.

### 4.3 Assessing Homophily Sensitivity Across Governing Equations

By training on seven datasets spanning homophily $h \in [0.11, 0.81]$ while holding other factors fixed, we reveal how variations in peak accuracy reflect each governing equation's sensitivity to structural homogeneity. To make this sensitivity explicit, we further introduce a post-hoc visualization.

**Accuracy–Homophily Response.** To assess how structural homophily affects each model's generalization ability, we fit a linear regression $A_{\mathrm{norm}}^{\star}(h) = \beta_0 + \beta_1 h$, where $A_{\mathrm{norm}}^{\star}(h)$ is the validation-optimal accuracy normalized per dataset via min–max scaling across models. This normalization removes absolute performance bias due to intrinsic dataset difficulty (e.g., some datasets lead to uniformly high or low accuracy), and highlights each model's relative standing on each dataset. As a result, the fitted slope $\beta_1$ more faithfully reflects a model's ranking as homophily varies.

As shown in Figure 3b, most diffusion-based models exhibit steep positive slopes: GCN and GAT yield $\beta_1 \approx 1.0$, confirming their strong dependence on homophilic structure. GRAND, fLode, and reaction–diffusion model ACMP show moderate positive correlations ($\beta_1 \approx 0.6$–$0.8$), indicating a moderate homophilic structural preference. Conservative wave models GraphCON and HamGNN also reach moderately high values ($\beta_1 \approx 0.7$), indicating that even energy-conserving dynamics still benefit from homophily. By contrast, KuramotoGNN is the only model with a negative slope (due to missing values). BRICK shows a flatter response curve with weak positive correlation, reflecting relative robustness to homophily.

TAKEAWAY. Acc–homophily slopes rank structural bias as: diffusion $\gg$ reaction $\approx$ wave/ Hamiltonian $>$ non-local $>$ oscillation. Diffusion requires homophilic graphs to excel, oscillation remains stable across $h$, and others show moderate dependence. Thus, most PDE-GNNs benefit from homophily, with even theoretically robust models exhibiting dataset-dependent behaviors in practice.

## 5 Discussion and Conclusion

This work presents PDEGNN-BENCH to systematically compare six families of PDE-governed GNNs under unified training and evaluation protocols. Our study uncovered how distinct governing equations shape depth scalability, oversmoothing behaviors, homophily sensitivity, and proposed a prior-driven, model-agnostic oversmoothing alarm. In the future, we will extend PDEGNN-BENCH to larger and more diverse real-world datasets to test the generality of our findings. Also, incorporating automated search over PDE-governed GNNs, combined with our oversmoothing alarm, may yield principled pipelines for designing depth-scalable and structure-robust models.

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

# A APPENDIX

## A.1 DEFINITION OF OVERSMOOTHING AND CATEGORIZING CRITERIA

### A.1.1 WHAT IS OVERSMOOTHING?

A message–passing layer in a GNN can be seen as a small dose of *blurring*: each node averages (or linearly mixes) its features with those of its neighbours. After a few layers, this denoises the signal, but if the blurring continues indefinitely, every pixel of the image turns the same grey. On a graph, every node within a connected component eventually receives (almost) the same embedding, so the model can no longer distinguish them. This loss of feature diversity is called *oversmoothing*.

*Preliminary.* Given an undirected weighted graph $G = (V, E, W)$ with $|V| = n$ and a node feature matrix $X^{(k)} \in \mathbb{R}^{n \times d}$ after $k$ message–passing layers, oversmoothing is informally defined as the degeneration of $X^{(k)}$ towards a low–dimensional (often constant) subspace as $k \to \infty$ (Li et al., 2018).

*Original formal definition* (Li et al., 2018): Linearising a Graph Convolutional Network (GCN) layer gives

$$H^{(k+1)} \;=\; \tilde{D}^{-1/2} \tilde{A} \tilde{D}^{-1/2} H^{(k)} W^{(k)}, \qquad \tilde{A} = A + I, \; \tilde{D}_{ii} = \sum_j \tilde{A}_{ij},$$

where $P = \tilde{D}^{-1/2} \tilde{A} \tilde{D}^{-1/2}$ is a symmetric Markov matrix with ordered eigenvalues $1 = \lambda_1 > \lambda_2 \geq \cdots \geq \lambda_n \geq -1$. If the weight matrices $W^{(k)}$ are well conditioned, the propagation dominates:

$$H^{(k)} \;\approx\; P^k H^{(0)} \;=\; \sum_{i=1}^{n} \lambda_i^k \, u_i u_i^\top H^{(0)}.$$

Because $|\lambda_i| < 1$ for $i > 1$, the high–frequency components $\lambda_i^k$ decay *exponentially* with depth. Hence $P^k \to u_1 u_1^\top$ ($u_1$ is the constant eigenvector), and

$$\left\| H_i^{(k)} - H_j^{(k)} \right\|_2 \;\longrightarrow\; 0, \qquad \forall \, i, j \text{ in the same component.}$$

Li et al. (2018) termed this depth regime **oversmoothing**. It is the inevitable outcome of repeatedly applying a smoothing (low–pass) operator on a connected graph: class–discriminative, high–frequency directions vanish exponentially fast, so deep GNNs collapse to a space with minimal class information. Recognising and *quantifying* this collapse, especially on graphs with different levels of homophily, has been an active research topic since the seminal analysis of Li et al. (2018).

While the phenomenon is easy to observe experimentally, its quantitative diagnosis is less straightforward. We propose a taxonomy based on the underlying *signal property* each metric monitors and on whether class label is required. A metric $\mathcal{M}(X)$ is assigned to **Category A–E** according to

- the **core quantity** whose decay or growth it tracks (energy, dispersion, capacity, class separability, or topological predictor), and
- its **label requirement**: label-free ($\circ$) vs. label-aware ($\bullet$).

*Notation.* Let $D$ be the degree matrix and $L = D - W$ the Laplacian. Denote by $y_i \in 1, \ldots, C$ the class label of node $i$ when available. For a vector $v$ we write $\|v\|_2$ for its Euclidean norm and $\sigma_i(X)$ for the $i$–th singular value of $X$.

### A.1.2 TAXONOMY OVER-SMOOTH CRITERIA

**A. Raw Smoothness / Energy Metrics ($\circ$): Dirichlet Energy (DE)**

$$\mathcal{E}(X) = \frac{1}{2} \sum_{(i,j) \in E} w_{ij} \|x_i - x_j\|_2^2 = \text{tr}(X^\top L X). \tag{1}$$

DE discretises the continuous Dirichlet integral $\int \|\nabla f\|^2$; minimising equation 1 therefore implements a graph heat flow (Oono & Suzuki, 2020). Deep GCNs drive $\mathcal{E}(X^{(k)}) \to 0$, signalling oversmoothing on *homophilous* graphs.

*Pros/Cons.* Easy $\mathcal{O}(|E|)$ computation, but blind to early class mixing on heterophilous graphs because $\mathcal{E}$ may remain non-zero while classes are already indistinguishable (Yan et al., 2022).

**B. Feature Dispersion Metrics (○): Mean Average Distance (MAD) and Total Pairwise Squared Distance (TPSD)**

A distance-normalized variant, MADGap, subtracts the average within a $K$–hop neighbourhood from the global MAD (Chen et al., 2020a).

$$\text{MAD}(X) = \frac{2}{n(n-1)} \sum_{i<j} \|x_i - x_j\|_2. \tag{2}$$

PairNorm (Zhao & Akoglu, 2019) proposed Total Pairwise Squared Distance (TPSD), which explicitly normalises embeddings so that TPSD stays constant, thereby preventing collapse.

$$\text{TPSD}(X) = \sum_{i<j} \|x_i - x_j\|_2^2 \tag{3}$$

*Pros/Cons.* Dispersion metrics capture any contraction but cannot tell benign contraction (necessary on heterophily) from harmful over-mixing.

**C. Capacity / Rank Collapse Metrics (○): Effective Rank and Spectral entropy**

Let $\sigma_i$ be the singular values of $X$. Define probabilities $p_i = \sigma_i / \sum_j \sigma_j$ and the Effective Rank is defined as

$$r_{\text{eff}}(X) = \exp\Big[-\sum_i p_i \log p_i\Big]. \tag{4}$$

If $r_{\text{eff}} \to 1$, the feature matrix becomes (almost) rank-one—a tight proxy for expressivity loss irrespective of homophily level (Zhang et al., 2025).

The Shannon entropy $H = -\sum_i p_i \log p_i$ itself can also be monitored (Yang et al., 2023).

*Pros/Cons.* Label–free and data–agnostic, but requires SVD ($\mathcal{O}(nd^2)$).

**D. Class–Aware Separability Metrics (●): Within/Between-Class Energy**

$$E_{\text{w}}(X) = \sum_{(i,j)\in E, y_i=y_j} \|x_i - x_j\|_2^2, \tag{5}$$

$$E_{\text{b}}(X) = \sum_{(i,j)\in E, y_i\neq y_j} \|x_i - x_j\|_2^2, \tag{6}$$

with ratio $\rho = E_{\text{w}}/E_{\text{b}}$ (Yan et al., 2022). Desired trends differ:

- *Homophily*: $E_{\text{w}} \downarrow$, $E_{\text{b}}$ stable ;⇒; $\rho \uparrow$.
- *Heterophily*: $E_{\text{w}}$ stable, $E_{\text{b}} \downarrow$ ;⇒; $\rho \downarrow$.

DDCD regularizes training to maximise the contrast $|E_{\text{w}} - E_{\text{b}}|$ and reports robustness across both regimes (Shen et al., 2024).

*Pros/Cons.* Decouples "collapse inside class" from "mixing across class", offering a unified lens. Needs labels and care on imbalanced data (add $\varepsilon$).

### A.1.3 METRIC-HOMOPHILY INTERACTION

A growing body of works does not treat graph homophily (or heterophily) as a mere dataset statistic, but rather as a *control variable* for analyzing why particular oversmoothing metrics succeed or fail.

**Node-Level Heterophily and Relative Degree**  Yan et al. (2022) defines $h_i$ as the fraction of edges from $i$ to different classes and shows that nodes with high $h_i$ and high relative degree oversmooth long before the Dirichlet Energy (Category A) approaches 0. This result exposes a "blind spot" of energy-based metrics on heterophilous regions of a graph.

**Rethinking Oversmoothing with effective rank**  This study (Zhang et al., 2025) presents convincing evidence that the effective rank of the feature matrix (CategoryC) drops in lock-step with test accuracy regardless of the homophily ratio (0.05–0.95 across 18 datasets). Energy and dispersion metrics fail on the strongly heterophilous subsets, confirming the robustness of rank-based measures.

**Dual-dimensional class-difference decoupling** By subtracting the within-class centroid from every embedding and maximising the between-class energy, DDCD (Shen et al., 2024) turns a Category D metric into a training regulariser. The method performs consistently on both Cora (homophilous) and Texas (heterophilous), highlighting the value of explicitly modelling class separability.

**Structural–Heterophily Ratio** Begga et al. (2023) proposes a *structural heterophily ratio*, which is the quotient of initial Laplacian energy to one-step "harmonicity" and derives closed-form bounds linking this ratio to accuracy degradation. Although not a metric of the embeddings themselves, these quantities guide scheduling of residual jumps or early stopping and are not metrics of $X$ themselves.

Collectively, these studies substantiate two claims: (i) label–agnostic metrics from Categories A and B become unreliable as heterophily rises, and (ii) rank–based (C) and class–aware (D) families maintain a near-monotonic relationship with accuracy across the entire homophily spectrum. Taken together, we ultimately adopt Category C (capacity/rank-based) and Category D (class-aware separability) metrics.

## A.2 DATASET DETAILS AND INFERENCE TIME

Table 3: Dataset description for node classification.

|  | **Texas** | **Wisconsin** | **Squirrel** | **Chameleon** | **Cornell** | **Citeseer** | **Cora** |
|---|---|---|---|---|---|---|---|
| **Hom. ratio $h$** | 0.11 | 0.21 | 0.22 | 0.23 | 0.3 | 0.57 | 0.81 |
| **# Nodes** | 183 | 251 | 5,201 | 2,277 | 183 | 3,327 | 2,708 |
| **# Edges** | 295 | 466 | 198,493 | 31,421 | 280 | 4,676 | 5,278 |
| **# Classes** | 5 | 5 | 5 | 5 | 5 | 7 | 6 |

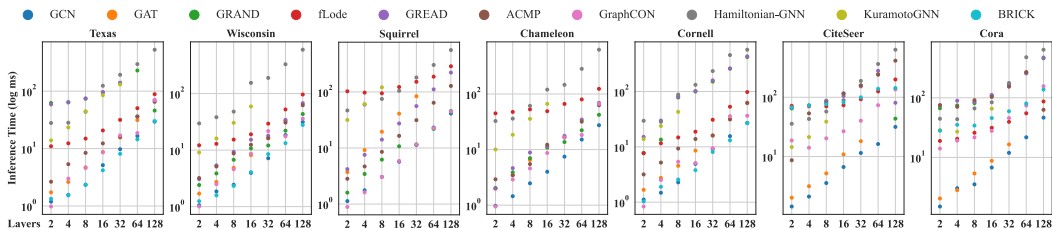

Figure 4: Inference Time

## A.3 VALIDATION-OPTIMAL CONFIGURATION FOR DATASET-MODEL PAIRS

Table 4 reports, for each model–dataset pair, the validation-optimal configuration $(\ell^\star, Acc_{val}^{(\ell^\star)}, r_{\text{eff}}^{(\ell^\star)}, S^{(\ell^\star)})$, which identifies the peak generalization layer $\ell^\star$ and its associated structural indicators.

Table 4: Validation-optimal configuration $(\ell^\star, Acc_{val}^{(\ell^\star)}, r_{\text{eff}}^{(\ell^\star)}, S^{(\ell^\star)})$ for each model–dataset pair.

|  | Texas | Wisconsin | Squirrel | Chameleon | Cornell | CiteSeer | Cora |
|---|---|---|---|---|---|---|---|
| GCN | (4, 0.59, 0.17, 0.29) | (4, 0.55, 0.16, 0.1) | (2, 0.28, 0.04, 0.02) | (2, 0.39, 0.06, 0.03) | (2, 0.55, 0.38, 0.11) | (2, 0.77, 0.03, 0.51) | (2, 0.88, 0.03, 0.58) |
| GAT | (4, 0.63, 0.35, 0.25) | (4, 0.58, 0.31, 0.32) | (2, 0.31, 0.3, 0.01) | (4, 0.46, 0.37, 0.16) | (2, 0.58, 0.61, 0.18) | (2, 0.77, 0.38, 0.41) | (2, 0.88, 0.4, 0.57) |
| GRAND | (2, 0.81, 0.92, 0.62) | (2, 0.85, 0.93, 0.59) | (32, 0.4, 0.91, 0.18) | (8, 0.58, 0.9, 0.31) | (2, 0.74, 0.94, 0.57) | (4, 0.77, 0.86, 0.34) | (8, 0.88, 0.78, 0.2) |
| fLode | (8, 0.72, 0.86, 0.21) | (4, 0.74, 1.01, 0.28) | (32, 0.58, 0.85, 0.01) | (4, 0.7, 0.91, 0.08) | (8, 0.67, 0.88, 0.15) | (2, 0.75, 0.88, 0.18) | (2, 0.86, 0.81, 0.3) |
| GraphCON | (2, 0.82, 0.29, 0.44) | (2, 0.83, 0.37, 0.6) | (2, 0.34, 0.53, 0.09) | (2, 0.46, 0.4, 0.22) | (2, 0.76, 0.37, 0.41) | (2, 0.77, 1.52, 0.41) | (2, 0.85, 1.6, 0.36) |
| GREAD | (2, 0.73, 0.92, 0.53) | (2, 0.82, 0.96, 0.58) | (64, 0.54, 0.91, 0.15) | (8, 0.65, 0.93, 0.3) | (2, 0.65, 0.94, 0.51) | (2, 0.76, 0.75, 0.41) | (8, 0.87, 0.66, 0.35) |
| ACMP | (2, 0.77, 0.83, 0.63) | (2, 0.81, 0.84, 0.52) | (32, 0.42, 0.93, 0.11) | (4, 0.6, 0.93, 0.23) | (4, 0.71, 0.88, 0.61) | (2, 0.77, 0.78, 0.4) | (8, 0.88, 0.57, 0.22) |
| Hamiltonian-GNN | (4, 0.63, 1.11, 0.12) | (2, 0.56, 1.11, 0.14) | (2, 0.55, 0.77, 0.79) | (2, 0.7, 0.62, 0.51) | (2, 0.55, 1.43, 0.25) | (2, 0.76, 0.89, 0.65) | (8, 0.87, 1.3, 0.78) |
| KuramotoGNN | (2, 0.73, 0.96, 0.48) | (2, 0.76, 1.0, 0.37) | (2, 0.44, 0.85, 0.26) | (2, 0.62, 0.85, 0.28) | (2, 0.58, 1.05, 0.38) | (2, 0.41, 1.1, 0.05) | (2, 0.37, 1.5, 16.09) |
| BRICK | (2, 0.87, 0.68, 0.65) | (4, 0.85, 1.07, 0.6) | (8, 0.53, 0.43, 0.11) | (4, 0.67, 0.29, 0.15) | (2, 0.8, 1.03, 0.52) | (4, 0.76, 0.62, 0.14) | (32, 0.88, 0.58, 0.53) |

## A.4 CONSTRUCTING THE $(r_{\text{eff}}, S)$ DENSITY ENVELOPES

**Step 1: Long-form tabulation.** Let $\mathcal{D}$ denote the set of datasets and $\mathcal{M}$ the set of models. For every $(d, m) \in \mathcal{D} \times \mathcal{M}$ we parse the table entry (l*, Acc, r_eff, S) into a record $(d, m, \ell^\star, \text{Acc}^\star, r_{\text{eff}}^\star, S^\star)$. Concatenating all records yields $\mathcal{R} = \{(r_{\text{eff},i}^\star, S_i^\star)\}_{i=1}^{|\mathcal{D}| \cdot |\mathcal{M}|}$.

**Step 2: Robust descriptive statistics.** For each dataset $d \in \mathcal{D}$ we collect $\mathcal{R}_d = \{(r_{\text{eff},i}^\star, S_i^\star) \mid \text{record } i \text{ belongs to } d\}$. Denote by $\hat{\mu}_r^{(d)} = \text{median}\{r_{\text{eff}}^\star \in \mathcal{R}_d\}$, $\hat{\mu}_S^{(d)} = \text{median}\{S^\star \in \mathcal{R}_d\}$ the coordinate-wise medians, and by $\widehat{\text{IQR}}_r^{(d)}$, $\widehat{\text{IQR}}_S^{(d)}$ the corresponding inter-quartile ranges. Following Tukey's normal approximation formula, a *robust* 95% *empirical interval* for each axis is

$$\mathcal{I}_r^{(d)} = \left[\hat{\mu}_r^{(d)} - \frac{1.57\widehat{\text{IQR}}_r^{(d)}}{\sqrt{N_d}}, \ \hat{\mu}_r^{(d)} + \frac{1.57\widehat{\text{IQR}}_r^{(d)}}{\sqrt{N_d}}\right], \qquad \mathcal{I}_S^{(d)} = \left[\hat{\mu}_S^{(d)} - \frac{1.57\widehat{\text{IQR}}_S^{(d)}}{\sqrt{N_d}}, \ \hat{\mu}_S^{(d)} + \frac{1.57\widehat{\text{IQR}}_S^{(d)}}{\sqrt{N_d}}\right],$$

where $N_d = |\mathcal{R}_d|$.

**Step 3: Kernel density estimation (KDE).** To visualise the joint distribution, we perform a bivariate Gaussian KDE $\hat{f}_d(r, S)$ over $\mathcal{R}_d$. We draw filled contour levels $\{(r, S) : \hat{f}_d(r, S) = \alpha_k\}_{k=1}^{K}$ with decreasing thresholds $\alpha_1 > \alpha_2 > \cdots > \alpha_K > 0$, producing a blue-scale *density heatmap*. Empirically, $K = 4$ and a bandwidth selected by Silverman's rule provide smooth yet localised contours.

**Step 4: Envelope overlay and export.** The Cartesian product $\mathcal{E}_d = \mathcal{I}_r^{(d)} \times \mathcal{I}_S^{(d)}$ is drawn as two pairs of dashed lines (red for $r_{\text{eff}}$, green for $S$).

---

**Algorithm 1** Envelope–guided oversmooth alarm (`Wisconsin`)

---

**Require:** training graph of the WISCONSIN dataset;
  1: maximum depth $L_{\max}$; history length $k$
**Ensure:** selected depth $\ell^\star$
  2: **Prior envelope for Wisconsin** (derived in Figure 3a):

$$\mathcal{I}_r^{(\text{WI})} = [0.69, \ 1.2], \quad \mathcal{I}_S^{(\text{WI})} = [0.3, \ 0.59].$$

  3: Initialize history buffers $H_r \leftarrow [\,]$, $H_S \leftarrow [\,]$.
  4: **for** $\ell \leftarrow 1$ **to** $L_{\max}$ **do**
  5:      Train (or load) the $\ell$-th layer and compute $(r_{\text{eff}}^{(\ell)}, S^{(\ell)})$.
  6:      Append $r_{\text{eff}}^{(\ell)}$ to $H_r$ and $S^{(\ell)}$ to $H_S$.
  7:      **if** $|H_r| \geq k$ **then**
  8:         $\text{trend}_r \leftarrow (\Delta H_r < 0 \text{ for last } k \text{ steps})$
  9:         $\text{trend}_S \leftarrow (\Delta H_S < 0 \text{ for last } k \text{ steps})$
10:         $\text{inside} \leftarrow (r_{\text{eff}}^{(\ell)} \in \mathcal{I}_r^{(\text{WI})}) \wedge (S^{(\ell)} \in \mathcal{I}_S^{(\text{WI})})$
11:         **if not** inside **and** $\text{trend}_r$ **and** $\text{trend}_S$ **then**
12:            **return** $\ell^\star \leftarrow \ell - 1$          ▷ early stop: oversmoothing detected
13:         **end if**
14:      **end if**
15: **end for**
16: **return** $\ell^\star \leftarrow L_{\max}$          ▷ no oversmoothing observed

---

## A.5 USE OF LARGE LANGUAGE MODELS (LLMs)

LLMs are only used as grammar checking and polishing of sentences.

