# OpenReview forum: "Towards Principled Design for Graph Neural Networks Through Governing Law of Dynamic Learning Behavior"
_ICLR.cc/2026/Conference — ICLR 2026 Conference Withdrawn Submission_

### Official Review · Reviewer_82R6 · 2025-10-15

**Soundness:** 1
**Presentation:** 2
**Contribution:** 1
**Rating:** 0
**Confidence:** 5

**Summary:**

This paper provides a summary of over-smoothing performance for several types of PDE-based GNN models.

**Strengths:**

A summary of some of the major PDE-based GNN models under two over-smoothing metrics is presented in a single paper. This can serve as a reference for future researchers.

**Weaknesses:**

1. The benchmarking should ideally focus on **mechanisms** (not classes of PDEs) that are specifically designed to mitigate over-smoothing, with insights or explanations of why they work or do not work, or under what situations they work best or collapse. This is not done in this work. For example, over-smoothing mitigation mechanisms like those in DREW, FROND, $\omega$GNN, or other recent models that do try to address over-smoothing should be analyzed and discussed. The paper claims to evaluate leading GNN models (I assume in the over-smoothing mitigation domain) but falls short of this objective.

1. Although the paper summarizes the over-smoothing mitigation performance of several classes of PDE-informed GNNs, it has not provided good insights and understanding of the effectiveness of over-smoothing mechanisms. Over-smoothing mitigation mechanisms must be designed to specifically target this phenomenon, which is inherent in all message-passing aggregation type of GNNs. Some of the GNN models presented are **not** suitable to mitigate over-smoothing (based on our current understanding of GNNs, not on what is claimed at the time the works are published, a hint is to look for theoretical guarantees): these models are more helpful for other aspects, e.g., some mitigates the heterophily issue or different geometries. To present these models in this over-smoothing benchmarking context does not provide the reader any significant insights into what works for mitigating over-smoothing and may in fact mislead new researchers into thinking that these models are not effective, when in fact they are good for targeting a different issue. There is a possibility to integrate the over-smoothing mechanisms in W1 into a model to obtain a model that mitigates over-smoothing **AND** other issues. It would be interesting to see what works and what not.

1. The novelty of this work is lacking as it is benchmark proposal. Furthermore, the metrics used are not new and taken from existing literature.

**Questions:**

1. It is unclear if the benchmarking has been done with the best hyperparameter and other options like neural network architecture choices in each of the PDE neural GNN models. The results shown are typically worse than those reported in the other papers, except for BRICK.

1. The work "Let Brain Rhythm Shape Machine Intelligence for Connecting Dots on Graphs" (the BRICK model) has not been published as at the time of this review (15 Oct 2025), nor is it available on any public repository like arXiv on and before 15 Oct 2025. This is a serious authorship anonymity leakage.

---

### Official Review · Reviewer_otPJ · 2025-10-16

**Soundness:** 3
**Presentation:** 3
**Contribution:** 3
**Rating:** 6
**Confidence:** 2

**Summary:**

This paper introduces PDEGNN-BENCH, a comprehensive benchmark designed to study GNNs inspired by Partial Differential Equations. Unlike prior empirical benchmarks, it grounds GNN evaluation in dynamical systems theory by linking each architecture to a corresponding governing equation—covering six canonical forms: isotropic/anisotropic diffusion, non-local diffusion, reaction–diffusion, Hamiltonian, wave, and oscillatory synchronization dynamics. Using seven datasets spanning a homophily range of 0.11–0.81, the framework investigates two key behaviors: oversmoothing (depth-induced representation collapse) and homophily sensitivity (topology-induced performance variance). New diagnostics—effective rank and class-mix score —quantify these effects. The study yields a model-agnostic oversmoothing alarm and regression-based homophily sensitivity index. The benchmark reveals distinct scaling and robustness patterns across PDE families and provides a theoretical lens for principled GNN design

**Strengths:**

1. Clear and straightforward writing. The article's argument is well-defined.
2. Six governing PDE classes offer a structured, interpretable organization of existing PDE-GNNs.
3. Provides a dataset-level prior requiring no label tuning, aligning with the paper’s “principled design” goal.
4. Provides a theoretical bridge between continuous dynamics and discrete message-passing mechanisms, supporting principled model design.

**Weaknesses:**

1. All benchmarks rely on small citation/social graphs, lacking evaluation on larger or dynamic graphs.
2. No direct evidence found for runtime or memory profiling (Appx A.2 Figure 4 mentions inference time but lacks analysis).
3. While the envelope heuristic (Fig. 3a) is visually convincing, there is no metric quantifying alarm accuracy or depth-stop saving
4. No direct evidence found for performance improvement when the alarm is applied during training.
5. PDE-governed models (especially non-local and Hamiltonian types) may entail higher complexity, yet wall-clock or GPU-memory cost is not reported (No direct evidence found).
6. No code has been provided for readers to reproduce.
7. Error in line 267: E_W -> E_b.

**Questions:**

Reference Weakness.

---

### Official Review · Reviewer_RY3x · 2025-10-30

**Soundness:** 2
**Presentation:** 2
**Contribution:** 2
**Rating:** 2
**Confidence:** 4

**Summary:**

This paper introduces PDEGNN-BENCH, a benchmark framework for systematically evaluating Graph Neural Networks governed by different partial differential equations (PDEs). The authors compare six PDE families (diffusion, non-local diffusion, reaction-diffusion, Hamiltonian, wave transport, oscillatory synchronization) across seven datasets with varying homophily levels (h ∈ [0.11, 0.81]). The paper proposes two main contributions: (1) a dual-metric oversmoothing detection system using effective rank and class-mix score, and (2) a model-agnostic oversmoothing alarm based on dataset-specific envelopes derived from validation optima. The authors claim to reveal distinct oversmoothing behaviors and homophily sensitivities across PDE families, providing guidance for principled GNN design.

**Strengths:**

1. Novel systematic comparison framework across PDE families fills an important gap in the literature.

2. Comprehensive experimental design covers representative models from each PDE family with unified evaluation protocols.

3. Generally well-written with clear problem motivation and systematic presentation of results.

4. The systematic comparison could provide valuable guidance for practitioners, and the oversmoothing detection tools may have practical utility despite methodological limitations.

**Weaknesses:**

1. Experimental Scope Limitations:
  - Only 7 datasets, heavily biased toward citation networks
  - Missing key application domains (molecular graphs, social networks) where PDE interpretations have physical meaning
  - No synthetic graphs with controlled properties to isolate specific effects
  - Insufficient diversity to support broad claims about "principled design"

2. Methodological Issues:
  - Oversmoothing alarm relies on circular definition using existing model optima
  - No validation that envelope method actually predicts oversmoothing vs. mere difference
  - Homophily analysis conflates multiple dataset properties (size, domain, density)
  - Integration time ≠ discrete layers equivalence unvalidated

3. Statistical Rigor:
  - No significance testing for performance differences
  - Linear regression with only 7 data points lacks power
  - Missing confidence intervals and effect size analysis
  - No correction for multiple comparisons across models

4. Technical Implementation:
  - Limited hyperparameter search (only 3 learning rates)
  - Missing crucial implementation details for reproducibility

5. Theoretical Depth:
  - Lacks rigorous theoretical analysis connecting PDE properties to observed behaviors
  - Explanations remain largely post-hoc empirical observations
  - No predictive framework for selecting appropriate PDE families

**Questions:**

1. Oversmoothing Alarm Validation:
  - How does the envelope method perform on architectures not included in the envelope construction?
  - Can you provide validation against ground-truth oversmoothing scenarios?
  - What is the false positive rate when the alarm stops beneficial training?

2. Depth Equivalence:
  - Have you empirically validated that integration time T corresponds to T discrete layers?
  - How do different ODE solvers affect your conclusions?
  - Why fix the step size at 1.0 rather than optimize it?

3. Homophily Analysis:
  - How do you disentangle homophily effects from confounding variables (size, domain, density)?
  - Why assume linear relationships rather than testing for non-linear effects?
  - Can you replicate findings on synthetic graphs with controlled homophily?

4. Statistical Significance:
  - Are the observed performance differences statistically significant?
  - What are the confidence intervals for the β₁ slopes in Figure 3b?
  - How does multiple testing correction affect your conclusions?

5. Generalization:
  - How confident are you that conclusions from citation networks generalize to other domains?
  - Would results hold on larger graphs (>10K nodes)?
  - Can you predict optimal PDE choice given graph properties?

6. Implementation Details:
  - How were "default hyperparameters" chosen for each model?
  - What are the computational cost differences across PDE families?

---

### Official Review · Reviewer_PV8N · 2025-10-31

**Soundness:** 3
**Presentation:** 3
**Contribution:** 3
**Rating:** 6
**Confidence:** 4

**Summary:**

This paper introduces PDEGNN-BENCH, a benchmark for evaluating GNNs governed by differential equations. It links physical dynamics (e.g., diffusion, wave, Hamiltonian) with learning behaviors like oversmoothing and homophily sensitivity. Six PDE-based GNN families are analyzed using two interpretable metrics—Effective Rank and Class-Mix Score—to measure expressiveness and separability. A model-agnostic oversmoothing alarm based on their joint statistical envelope is proposed. Experiments on seven datasets reveal how different governing equations influence depth scalability, stability, and structural bias.

**Strengths:**

1. A systematic, cross-equation comparison of PDE-based GNNs under unified conditions.

2. A novel, model-agnostic "oversmoothing alarm" that can warn during training when a model is becoming too deep and losing expressive power.

3. A quantitative assessment of how sensitive each PDE-based GNN is to the homophily level of the graph data.

**Weaknesses:**

1. Despite rich analysis, the evaluation relies mainly on small datasets; assessment on much larger and more diverse graphs is needed to substantiate the conclusions.

2. The set of ODE models is limited; additional baselines (e.g., [1], [2]) should be included.

3. How sensitive is oversmoothing to the choice of numerical solver in PDE-GNNs (e.g., Euler, RK4, adaptive, Symplectic Euler)? Do solver-induced numerical dissipation/dispersion effects meaningfully alter (r_eff, S)?

4. The Hamiltonian GNN in HamGNN does not strictly follow canonical Hamiltonian equations due to multiple layers. Would it be more appropriate to evaluate HANG from [2]?

[1] Q. Kang et al., “Unleashing the potential of fractional calculus in graph neural networks with FROND,” ICLR 2024.

[2] Zhao, K., Kang, Q., Song, Y., She, R., Wang, S., & Tay, W. P. (2023). Adversarial robustness in graph neural networks: A Hamiltonian approach. NeurIPS 36, 3338–3361.

**Questions:**

See the weakness section

---

### Note · Authors · 2025-11-12

I have read and agree with the venue's withdrawal policy on behalf of myself and my co-authors.